# Cross-Contamination Risk of Dental Tray Adhesives: An In Vitro Study

**DOI:** 10.3390/ma14206138

**Published:** 2021-10-15

**Authors:** Isabel Paczkowski, Catalina S. Stingu, Sebastian Hahnel, Angelika Rauch, Oliver Schierz

**Affiliations:** 1Department of Prosthodontics and Materials Science, University of Leipzig, Liebigstrasse 12, D-04103 Leipzig, Germany; Isabel.paczkowski@medizin.uni-leipzig.de (I.P.); sebastian.hahnel@medizin.uni-leipzig.de (S.H.); angelika.rauch@medizin.uni-leipzig.de (A.R.); 2Institute for Medical Microbiology and Virology, University Hospital of Leipzig, Liebigstrasse 21, D-04103 Leipzig, Germany; catalinasuzana.stingu@medizin.uni-leipzig.de

**Keywords:** dental tray adhesive, reusable brush, disinfectant additive, cross-contamination risk, disinfection

## Abstract

Background: The aim of this study was to investigate the risk of cross-contamination in dental tray adhesives with reusable brush systems. Methods: Four dental tray adhesives with different disinfectant components were examined for risk as a potential transmission medium for *Staphylococcus aureus*, *Escherichia coli*, *Pseudomonas aeruginosa*, *Streptococcus oralis*, and *Candida albicans*. Bacterial and fungal strains were mixed with artificial saliva. The contaminated saliva was intentionally added to tray adhesive liquid samples. At baseline and up to 60 min, 100 microliters of each sample were collected and cultivated aerobically on Columbia and Sabouraud agar for 24 or 48 h, respectively. Results: At baseline, contamination with *Staphylococcus aureus* and *Candida albicans* could be identified in three out of four adhesives. In the subsequent samples, low counts of up to 20 colony-forming units per milliliter could be observed for *Staphylococcus aureus*. All other strains did not form colonies at baseline or subsequently. Adhesives with isopropanol or ethyl acetate as disinfectant additives were most effective in preventing contamination, while adhesives with hydrogen chloride or acetone as a disinfectant additive were the least effective. Conclusion: Within 15 min, the tested adhesives appeared to be sufficiently bactericidal and fungicidal against all microorganisms tested.

## 1. Introduction

Numerous guidelines and hygiene recommendations outline proper aseptic handling and corresponding workflows in everyday dentistry [1,2], which not only protect patients but also ensure workplace safety for medical healthcare providers [3]. In recent years, disposable products have gained importance, whereas reusable materials have become less frequently used in direct patient contact. However, monetary and ecological aspects play a relevant role in the decision-making process. Therefore, reusable materials may stay relevant in routine dental practice [4].

Since approximately 1.2 million impressions are billed annually in Germany alone [5], conventional impression-taking is still state of the art despite the availability of digital impression-taking procedures [6,7]. Impression tray adhesives provide a chemical adhesion of impression materials to the tray, prevent distortion, and ensure dimensional stability of the impression after removal from the mouth. The adhesive is usually delivered in a reusable glass flask with a screw cap. On the inside of the cap, a brush is fixed for applying the adhesive liquid. The use of the brush may lead to contamination of the adhesive reservoir in the glass flask if there is no proper intermediate disinfection of the impression tray after intraoral try-in. Lasting contamination of the reservoir could pose a risk to all subsequently treated patients [3,8,9,10]. This would expose risk to patients who are suffering from severe primary disease and immunosuppression as well as the increasing number of multimorbid elderly. Exacerbating this issue, the current COVID-19 pandemic has further underlined the relevance of proper hygiene measures.

Manufacturers assume sufficient disinfectant activity through additives such as isopropanol, ethyl acetate, hydrogen chloride, acetone, toluene, or trichloroethane. The first scientific considerations addressing the risk of cross-contamination in the impression-taking process were published in 1987 [9]. Six years later, the disinfectant effects of different tray adhesives in three in vitro cultured bacterial strains (*Staphylococcus aureus*, *Salmonella* Choleraesuis, and *Pseudomonas aeruginosa*) were investigated. Only the Express adhesive, with additives trichloroethane and toluene, showed small deficits in antibacterial effect [8]. In the recent literature, a publication contradicted the hypothesis that adhesives disinfect sufficiently [10]. None of the adhesive systems tested revealed sufficient disinfectant activity when using the Kirby–Bauer zone of inhibition method. Apart from some in vitro bacterial strains (*Pseudomonas aeruginosa*, *Escherichia coli*, *Streptococcus mutans,* and *Staphylococcus aureus*), the study also investigated bacterial cultures from twenty saliva samples. Driven by these results, the contamination of an impression tray adhesive in glass flasks with repeated-use brushes was investigated under clinical conditions. While no quantitative analysis was performed, the qualitative analysis showed bacterial contamination in 6 out of 400 agar plates [11].

Against this background, the current in vitro study aimed to observe the disinfecting effect of four commercially available tray adhesives with reusable brush systems that had been deliberately contaminated with potentially pathogenic bacteria and fungi of the oral microbiome. The null hypothesis was that no microorganisms could be cultivated in the dental impression tray adhesive liquid. 

## 2. Materials and Methods 

Four common adhesive systems with different disinfectant additives were investigated, including an adhesive with the disinfecting additive isopropanol (FA: Fix Adhesive; Dentsply DeTrey GmbH, Konstanz, Germany; charge: 2001000870/1905000723); an adhesive with ethyl acetate (UA: Universal Adhesive; Kulzer GmbH, Hanau, Germany; charge: K01005-4/-8/-6), an adhesive with hydrogen chloride, isopropanol, and ethyl acetate (PA: Polyether Adhesive; 3M GmbH, Neuss, Germany; charge: 5386594); and one with ethyl acetate and acetone (PCTA: Polyether Contact Tray Adhesive; 3M GmbH, Neuss, Germany; charge: 4581863) (Figure 1). All adhesives were tested for sterility before use by inoculating the tested adhesive liquid onto Columbia and Chocolate agar and examining the agar plates after a 24 h incubation time.

Clinically common and potentially pathogenic bacteria and fungi (freeze-dried bacterial and fungal strains from American Type Culture Collection (ATCC) and German Collection of Microorganism and Cell Cultures (DSM)) were selected as test strains, including 

Staphylococcus aureus (ATCC 29213);Escherichia coli (ATCC 25922);Pseudomonas aeruginosa (ATCC 29213);Streptococcus oralis (DSM 20627);Candida albicans (ATCC 90028).

Reference strains were cultivated aerobically on Columbia agar for 24 h and on Sabouraud agar for 48 h. Artificial saliva was prepared in the laboratory according to the recipe of Rosentritt et al. [12,13] and stored in a refrigerator at –20 degrees Celsius (°C). Prior to use, the artificial saliva was brought to room temperature and tested for sterility. In order to verify the sterility, 100 microliters (µL) of the saliva was placed onto Columbia and Chocolate agar and examined after an incubation time of 24 h.

Growing colonies of the reference strains were isolated and added to the artificial saliva in a starting concentration of 1 × 10^9^ for bacteria and 1 × 10^5^ colony-forming units per milliliter (CFU/mL) for fungi according to the average occurrence of bacteria and fungi in the oral cavity [14,15,16]. The bacterial count was photometrically verified by three subsequent measurements using an optical density of 0.85 for bacteria and 0.125 for fungi at a wavelength of 580 nanometers (Ultraspec 2000 UV-VIS spectrophotometer, Pharmacia Biotech, Waldkirch, Germany). The fungal strain was diluted 1:100 in order to obtain a final fungal concentration of 1 × 10^5^ CFU/mL. 

Prior to initiating the growth inhibition test, the contaminated saliva samples were examined regarding bacterial and fungal purity. The purity was verified by inoculating the samples onto agar plates. After incubation, the plates were visually inspected, and the colonies were identified using the matrix-assisted laser desorption–ionization time-of-flight mass spectrometry (MALDI-TOF; VITEK^®^ MS, bioMérieux, Lyon, France). Twenty microliters of the contaminated saliva was added to 2 mL of the respective adhesive liquid (ratio of 1:100) and mixed for five seconds (IKA VF2 Vortex Mixer, IKA^®^-Werke GmbH & Co. KG, Staufen, Germany). Twenty microliters corresponds to the average amount of saliva adhering to an impression tray after try-in. This amount was determined by using a precision scale (Cubis^®^, Sartorius AG, Goettingen, Germany) and 20 impression tray samples. 

At baseline and in 15 min intervals up to 60 min, 100 µL of each sample was inoculated onto Columbia and Sabouraud agar using a pipette system (Multipette^®^ (4780)); Eppendorf Combitips advanced^®^, Eppendorf AG, Hamburg, Germany) and a sterile disposable spatula. The agar plates were incubated aerobically for 24 or 48 h at 37 °C and 5 percent (%) CO_2_ (Heracell 150i CO_2_ Incubator, Thermo Fisher Scientific, Dreieich, Germany), and the bacterial count was documented (Figure 2). 

Initially, 10 samples per bacterium or fungus in combination with each adhesive were examined (5 strains × 4 adhesives × 10 test rows × 5 timeslots). Due to a relevant number of positive results after the initial test series, *Staphylococcus aureus* was tested with a further 10 samples to allow statistical demarcation between the various adhesives. In total, 1200 agar plates were screened.

The counting was repeated three times for an exact determination of the bacterial or fungal count, and the results were averaged. In addition, agar plates with a bacterial count of more than 50 colonies were divided into quarters, more than 100 colonies into eighths, and more than 200 colonies into sixteen parts to facilitate the counting process. If the number of colonies exceeded 300, proper counting was no longer possible. These counts were defined as “confluent culture”. For statistical evaluation, confluent cultures were included with 300 CFU per agar. 

The statistical software package STATA was used for descriptive analysis and statistical evaluation of the results (Stata Statistical Software: Release 15.1. StataCorp LP, College Station, TX, USA). The Wilcoxon rank-sum test and the Kruskal–Wallis test were performed for statistical analysis. Level of significance was set to *p* = 0.05, and for compensation of multiple testing, Bonferroni correction was applied.

## 3. Results

At baseline, in three out of four adhesives (UA, PA, PCTA), positive bacterial growth of Staphylococcus aureus was detected. The bacterial count varied significantly depending on the examined adhesive, with PA and PCTA showing the greatest deficits in instant disinfectant efficiency, allowing bacterial growth on all agar plates (100%). UA showed growth of Staphylococcus aureus in 65 % of all samples. FA allowed no growth at all (Figure 3). 

In 75% of the PCTA samples, confluent cultures of Staphylococcus aureus were detected. Additionally, fungal growth was identified in 5% of PA cultures. Except for PA, all adhesives inhibited fungal growth completely. A statistical significance could be proven when comparing the different adhesives at baseline using the Kruskal–Wallis test (*p* = 0.002). FA proved to have the best disinfectant properties compared to all other tested adhesives (Wilcoxon, all *p* = 0.001) for Staphylococcus aureus at baseline. UA’s disinfectant properties proved to be superior to PCTA (UA vs. PA *p* = 0.057; UA vs. PCTA *p* = 0.026) in Staphylococcus aureus. No statistically significant difference could be detected between PA and PCTA (*p* = 0.311).

After an incubation time of 15 min, 15% of PA showed growth of Staphylococcus aureus, with an average bacterial count of 13.3 CFU/mL (standard deviation of 4.71). No growth was identified for all other strains and adhesives. After a period of 30 min, 5% of PCTA showed Staphylococcus aureus counts of 10 CFU/mL (Figure 4, Table 1). No growth was identified on all other samples. No bacterial or fungal cultures were detected at either 45 or 60 min. 

## 4. Discussion

The bacterial and fungal cultures detected after an incubation of 15 min or longer were not clinically relevant since no more bacterial and fungal growth could be identified in the current study. However, compared to the initial bacteria count of 10^9^ CFU/mL in the contaminated saliva, bacteria were detected in 15% of PA samples after 15 min and in 5% of PCTA samples after 30 min. These samples revealed a small colony count of up to 20 CFU/mL. Therefore, the risk of cross-contamination with reusable brushes is highly unlikely, and the null hypothesis has to be accepted.

Intermediate disinfection of the impression trays after try-in seems unnecessary since the adhesives’ additives feature a sufficient disinfecting effect. However, it should be noted that significant differences exist in the disinfectant potency of the examined adhesives. In the current study, only FA could suppress any growth of bacteria and fungi due to its effective additive isopropanol. Isopropanol has an optimum bactericidal concentration between 60 and 90 % and can kill resistant *Staphylococcus aureus* within 10 s [17]. Even at baseline, no positive bacterial or fungal growth could be detected. PA and PCTA, which contain hydrogen chloride, isopropanol, acetone, and ethyl acetate as additives, showed the lowest antimicrobial effect. Different statements regarding the disinfectant efficiency of tray adhesives have led to increasing insecurities about reusable adhesive systems. In 1993, Herman [8] assumed a sufficient disinfectant effect of tray adhesives, while following publications by Pollak [10] and Schierz [11] contradicted the results and documented a potential cross-contamination risk for patients. The Kirby–Bauer method, as applied by Pollak [10], is to be evaluated critically, as it causes evaporation of the additives in the adhesive liquid, leading to a loss of disinfecting components and a corresponding distortion of test results. In addition, the specified amount of adhesive and saliva is not clinically relevant, which explains why the procedures do not allow any practical conclusions for a dental practice. Schierz et al. documented viable bacteria in 1.5% of investigated samples [11]; this should, however, be interpreted with caution, as several dermal bacteria were detected, and no quantification of the bacteria was performed.

In the current study, artificial saliva was combined with clinically relevant bacteria and fungi to optimize the informative value. While natural saliva shows individual variations in bacteria quantity and species [18], artificial saliva is produced according to a fixed recipe, can be reproduced in sufficient quantities, and has consistent quality. In addition, the possibility of adding individual bacterial and fungal species to the artificial saliva, as shown in this study, can avoid the competition between them regarding nourishment and habitat [19,20,21], which allows a reliable statement and reproducible results. 

To guarantee sterile saliva, individual components were sterilized before merging. The mucin (Mucin from a porcine stomach; Sigma-Aldrich, St. Louis, MO, USA) was decontaminated according to the manufacturer’s recommendation by placing the powder in 95% ethanol and heating the covered mucin at 70 °C for 24 h. Phosphate-buffered saline (PBS; Sigma Aldrich, St. Louis, MO, USA) was filtered (0.2 µm) before use.

To simulate clinical conditions, common and potentially pathogenic bacteria and fungi were chosen for the present investigation. *Staphylococcus aureus* is known as the main pathogen for bacterial endocarditis and osteomyelitis [22,23,24,25]. *Escherichia coli* is the most frequent enteric intestinal bacterium [26]. *Pseudomonas aeruginosa* is a hospital pathogen with increasing resistance, responsible for severe pneumonia and persistent urinary tract infections [27]. *Streptococcus oralis* can be assumed as a reference resident bacterium of the oral microflora. *Candida albicans* was included as the most common fungus in the oral environment and trigger of candidiasis [28,29].

The strengths of this study include the reproducibility of the testing approach, the clinically relevant chosen observation time with 15 min intervals, and the inclusion of common pathogens. However, the bacterial and fungal selection was not completely representative, and—particularly in the contemporary pandemic context—viruses should also be subjects of further investigation [30]. Within a period of 15 min, all products showed a sufficient disinfectant effect. Using reusable brush systems in adhesive systems is less likely to create a critical risk for patients due to contamination of the adhesive reservoir with the tested bacteria and fungi.

The current COVID-19 pandemic has underlined the relevance of proper hygiene standards and has increased awareness regarding adequate protective equipment in everyday dentistry. The use of disposable utensils to safeguard dental professionals and patients has gained in importance. Companies offer alternative forms of application to minimize transmission risks, such as single-use brush systems or adhesive liquid in spray form. However, economic and environmental aspects also play a relevant role. Further studies concerning potential cross-contamination risk should include viruses.

## 5. Conclusions

The tested impression tray adhesives and the corresponding additives appear to be sufficiently bactericidal and fungicidal. Since only a low count of *Staphylococcus aureus*, up to 20 CFU/mL, could be identified after baseline, the cross-contamination risk among patients is extremely low. Furthermore, compared to the initial bacteria count of 10^9^ CFU/mL, the remaining amount of 20 CFU/mL proves the tray adhesives’ extremely high disinfectant capacity. 

The pandemic has provoked an increasing awareness in patients regarding possible transmissions of microorganisms and hygiene standards. This investigation underlines that the clinical use of the tested tray adhesives is safe. 

## Figures and Tables

**Figure 1 materials-14-06138-f001:**
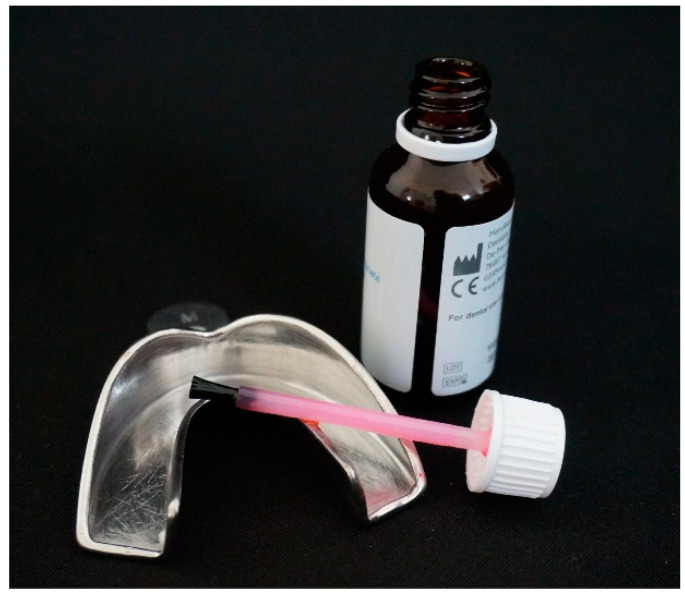
Impression tray, reusable brush, and adhesive flask (sample).

**Figure 2 materials-14-06138-f002:**
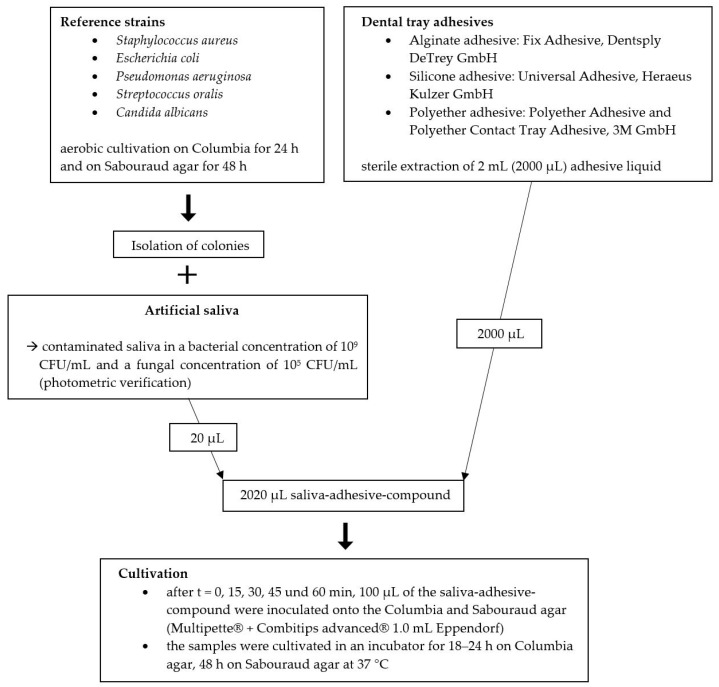
Overview of the materials and method.

**Figure 3 materials-14-06138-f003:**
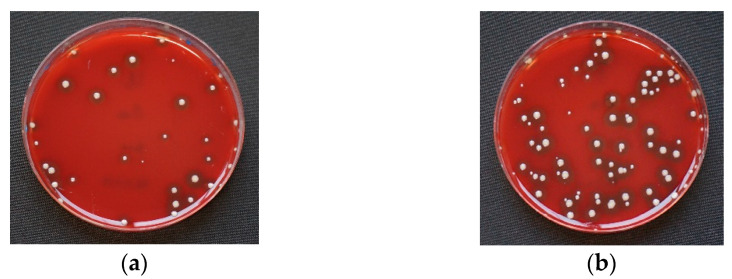
Examples for *Staphylococcus aureus* on Columbia agar at baseline after 0 min incubation. (**a**) Universal Adhesive (UA); (**b**) Polyether Adhesive (PA); (**c**) Polyether Contact Tray Adhesive (PCTA). In Fix adhesive (FA), no colonies of *Staphylococcus aureus* could be detected.

**Figure 4 materials-14-06138-f004:**
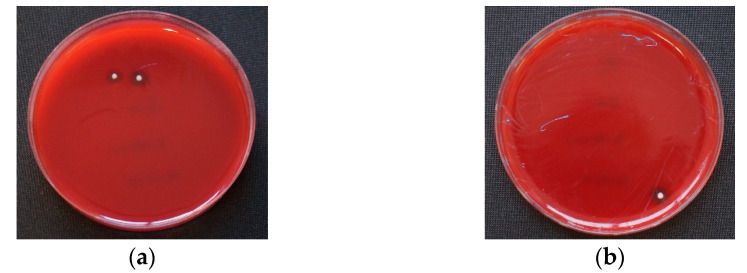
Examples of *Staphylococcus aureus* on Columbia agar. (**a**) Polyether Adhesive (PA) after 15 min incubation; (**b**) Polyether Contact Tray Adhesive (PCTA) after 30 min incubation. In Universal Adhesive (UA) and Fix Adhesive (FA), no colonies could be detected at both times.

**Table 1 materials-14-06138-t001:** Agar probes of *Staphylococcus aureus* and *Candida albicans* up to 30 min incubation. All other strains showed no viable bacteria or fungi at any time.

Microorganism	t_0_		t_15_		t_30_		t_0_ vs. t_15_	t_15_ vs. t_30_
	Median in CFU/mL	Min; Max	Medianin CFU/mL	Min; Max	Median in CFU/mL	Min; Max	*p*-Value ^1^	*p*-Value ^1^
*Staphylococcus aureus*								
Fix	0	0; 0	0	0; 0	0	0; 0	n.a.	n.a.
Universal	15	0; 1887	0	0; 0	0	0; 0	<0.001	<0.001
Polyether	328	10; 2223	0	0; 20	0	0; 0	<0.001	<0.001
Polyether Contact Tray	3000	220; 3000	0	0; 0	0	0; 10	<0.001	<0.001
*Candida albicans*								
Fix	0	0; 0	0	0; 0	0	0; 0	n.a.	n.a.
Universal	0	0; 0	0	0; 0	0	0; 0	n.a.	n.a.
Polyether	0	0; 10	0	0; 0	0	0; 0	0.317	n.a.
Polyether Contact Tray	0	0; 0	0	0; 0	0	0; 0	n.a.	n.a.

^1^ Wilcoxon rank-sum test; n.a. = not applicable.

## Data Availability

Data available on request.

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
