# Peer review of "Cross-Contamination Risk of Dental Tray Adhesives: An In Vitro Study"

_materials, 2021, doi:10.3390/ma14206138_

Round 1
Reviewer 1 Report
This paper investigated the risk of cross-contamination in dental tray adhesives. The authors systematically studied four dental tray adhesives and examined their potential transmission to five medium. They also evaluated the performance of different solvents as disinfectant additive. Overall, I think this paper was well written. The experiments were well designed and the results were clearly presented. I really like the introduction section where the historical efforts of this area and the limitations of the current available work were stated in detail. The scope and content of this paper fit Materials very well. I would recommend this paper for publication after minor revisions. Below are my specific comments:
- The authors need to go back and read through the paper in detail to identify grammar mistakes. Take the first sentence of the abstract as an example, which is not a complete sentence. There are many small mistakes throughout the paper and I think it's necessary to correct them before publication.
- Figure 2 has low resolution and the font size is very small. Please address this.
- Personally, I found Table 1 one a little crowded with too much information. For example, after each median value, the authors also include the (1.; 3. Quartile). To me, the quartile values are not very important, especially they were not discussed in the manuscript. I would consider removing this information.
Author Response
We thank the reviewers for their constructive comments, which helped us to improve the quality of the manuscript considerably. We addressed all comments and marking relevant changes in yellow within the manuscript; the detailed replies follow:
- The authors need to go back and read through the paper in detail to identify grammar mistakes. Take the first sentence of the abstract as an example, which is not a complete sentence. There are many small mistakes throughout the paper and I think it is necessary to correct them before publication.
Thank you for this valuable remark. We asked a professional English Editing Service to revise the article and I hope the linguistic improvement is perceptible.
We rephrased the first sentence of the abstract: “The aim of this study was to investigate the risk of cross-contamination in dental tray adhesives with reusable brush systems.”
- Figure 2 has low resolution and the font size is very small. Please address this.
We enlarged the font size of Figure 2 from 9 to 10 Pt. and we improved the resolution by transforming the figure into a high-resolution graphic.
- Personally, I found Table 1 one a little crowded with too much information. For example, after each median value, the authors also include the (1.; 3. Quartile). To me, the quartile values are not very important, especially they were not discussed in the manuscript. I would consider removing this information.
Thank you for the valuable proposal. I deleted the quartile values in Table 1. The table now allows a quicker and simplified overview.
Reviewer 2 Report
The title of manuscrit: Risk of Cross Contamination of Dental Tray Adhesives: In Vitro Study.
The aim of the study was to observe the disinfection effect of four commercially available tray adhesives with reusable brush systems, which were intentionally contaminated with potentially pathogenic bacteria and oral fungi and microbiota.
In my opinion, the purpose of the work has been precisely defined and the selected research methodology is adequate. The test results are clearly presented in the section.
Dear Authors,
I have a few comments to improve the quality of your work:
- The following strains were selected for the study (line nr 92-96):
- Staphylococcus aureus (ATCC 29213),
- Escherichia coli (ATCC 25922),
- Pseudomonas aeruginosa (ATCC 29213),
- Streptococcus oralis (DSM 20627) i
- Candida albicans (ATCC 90028).
However in section Conclusion in line 249 you confirm, ..”that viruses should corroborate the data of the current study”…In my opinion, based on the research carried out, no such conclusion can be drawn. I propose to remove it conclusion.
- In line nr 52-53 „The current COVID-19 pandemic has underlined the relevance of proper hygiene measures, too”. The proposed research does not concern the study of viruses, but only bacteria and fungi. I propose to delete or expand this statement to justify its inclusion in the manuscript.
- I propose to shorten the conclusions and present them more clearly.
- I propose to add a statement at the end of the discussion that the null hypothesis turned out to be correct and was supported by research results.
- Improving the English language especially in Introducion in lines 41-53.
Author Response
- The following strains were selected for the study (line nr 92-96):
Staphylococcus aureus (ATCC 29213),
Escherichia coli (ATCC 25922),
Pseudomonas aeruginosa (ATCC 29213),
Streptococcus oralis (DSM 20627),
Candida albicans (ATCC 90028).
However in section Conclusion in line 249 you confirm, ..”that viruses should corroborate the data of the current study”…In my opinion, based on the research carried out, no such conclusion can be drawn. I propose to remove it conclusion.
Thank you for the remark. We removed this statement.
- In line nr 52-53 „The current COVID-19 pandemic has underlined the relevance of proper hygiene measures, too”. The proposed research does not concern the study of viruses, but only bacteria and fungi. I propose to delete or expand this statement to justify its inclusion in the manuscript.
We extended this statement in the discussion part as recommended by reviewer #3 to justify its inclusion.
- I propose to shorten the conclusions and present them more clearly.
The conclusion appears to be more concise and abbreviated after deleting the statement in line 249 and following as recommended in comment #1.
- I propose to add a statement at the end of the discussion that the null hypothesis turned out to be correct and was supported by research results.
Thank you for the proposal. We modified the statement regarding the null hypotheses in the discussion part in line 178: “Therefore, the risk of cross-contamination by using reusable brushes is highly unlikely, and the null hypothesis has to be accepted”
- Improving the English language especially in Introduction in lines 41-53.
As mentioned in comment #1 of the first reviewer, we asked a professional English Editing Service to revise the article. I hope the linguistic improvement is perceptible.
Reviewer 3 Report
Dear authors, this is a well conducted study where the methodology is accurate and well designed.
You have mentioned in introduction, limitations and conclusions only a little about the pandemic situation and how this could influence the present research. I would comment the COVID-19 situation thoroughly and how this has affected the way in which we use certain materials at the clinic. Are these brushes still being used? Or have they changed them to one-use brushes? Are there any new protocols applied to this kind of brushes?
Please explain further in discussion
Author Response
- You have mentioned in introduction, limitations and conclusions only a little about the pandemic situation and how this could influence the present research. I would comment the COVID-19 situation thoroughly and
how this has affected the way in which we use certain materials at the clinic. Are these brushes still being used? Or have they changed them to one-use brushes? Are there any new protocols applied to this kind of brushes? Please explain further in discussion.
Thank you for the valuable proposal of improvement. We added a paragraph regarding the COVID-19 situation in the discussion (from lines 226 to 232).